# Genome-Wide DNA Methylation and Gene Expression Profiles in Cows Subjected to Different Stress Level as Assessed by Cortisol in Milk

**DOI:** 10.3390/genes11080850

**Published:** 2020-07-25

**Authors:** Marcello Del Corvo, Silvia Bongiorni, Bruno Stefanon, Sandy Sgorlon, Alessio Valentini, Paolo Ajmone Marsan, Giovanni Chillemi

**Affiliations:** 1Department of Animal Science Food and Nutrition—DIANA, Nutrigenomics and Proteomics Research Centre—PRONUTRIGEN, and Biodiversity and Ancient DNA Research Centre, Università Cattolica del Sacro Cuore, 29122 Piacenza, Italy; paolo.ajmone@unicatt.it; 2Istituto di Biologia e BiotecnologiaAgraria, Consiglio Nazionale delle Ricerche, 20133 Milan, Italy; 3Department of Ecological and Biological sciences DEB, University of Tuscia, 01100 Viterbo, Italy; bongiorni@unitus.it; 4Department of Agrifood, Environmental and Animal Science–University of Udine, 33100 Udine, Italy; bruno.stefanon@uniud.it (B.S.); sandy.sgorlon@uniud.it (S.S.); 5Department for Innovation in Biological, Agro-food and Forest systems DIBAF, University of Tuscia, 01100 Viterbo, Italy; alessio@unitus.it (A.V.); gchillemi@unitus.it (G.C.); 6Institute of Biomembranes, Bioenergetics and Molecular Biotechnologies, IBIOM, CNR, 70126 Bari, Italy

**Keywords:** cortisol secretion, DNA methylation, bisulfite sequencing, gene expression, dairy cattle health

## Abstract

Dairy cattle health, wellbeing and productivity are deeply affected by stress. Its influence on metabolism and immune response is well known, but the underlying epigenetic mechanisms require further investigation. In this study, we compared DNA methylation and gene expression signatures between two dairy cattle populations falling in the high- and low-variant tails of the distribution of milk cortisol concentration (MC), a neuroendocrine marker of stress in dairy cows. Reduced Representation Bisulfite Sequencing was used to obtain a methylation map from blood samples of these animals. The high and low groups exhibited similar amounts of methylated CpGs, while we found differences among non-CpG sites. Significant methylation changes were detected in 248 genes. We also identified significant fold differences in the expression of 324 genes. KEGG and Gene Ontology (GO) analysis showed that genes of both groups act together in several pathways, such as nervous system activity, immune regulatory functions and glucocorticoid metabolism. These preliminary results suggest that, in livestock, cortisol secretion could act as a trigger for epigenetic regulation and that peripheral changes in methylation can provide an insight into central nervous system functions.

## 1. Introduction

Dairy animals experience a large variety of stressors that can modify normal behavior and growth, leading to a decrease in productive performances. Normal physiological events such as calving, onset of lactation, lactation, weaning and group rearrangement can cause metabolic and environmental conditions which lead to stress, an impairment of animal wellbeing and a consequent decrease in the quality of animal products. Under these stressful conditions, the hypothalamic–pituitary–adrenal (HPA) axis, the autonomic nervous system, and the immune system are called into action to re-establish homeostasis [1]. Stress modifies the secretion of various hormones, as glucocorticoids, which play an important role in immunity [2,3,4]. A common feature of stress situations is an increase in cortisol secretion that produces a mobilization of energy reserves [5]. For example Bertulat et al. [6] showed higher concentration of glucocorticoid metabolites in the feces of drying off cows with higher milk yield while Horst and Jorgensen [7] reported an increase in plasma cortisol in cows with milk fever, associated with immune suppression and an increased risk of clinical mastitis and high somatic cell count in milk. To date, since blood sampling is often a source of stress, milk can be viewed as a viable and non-invasive way to measure cortisol levels in lactating cows, as it can be measured without the manipulation of animals.

Even though many methods of wellbeing evaluation are reported in the literature, cortisol is still considered one of the gold standards to describe animal response to stress [8]. However, until now, no systematic studies have investigated the epigenetic mechanisms related to cortisol secretion during stress conditions in lactating cows.

Epigenetics are defined as heritable changes in gene activity and expression and consequently in phenotype, which occur without alteration in the underlying DNA sequence [9]. At least four molecular systems, including DNA methylation [10], histone post-translational modifications [11], non-coding RNAs [12] and chromatin remodeling [13], control gene expression. The epigenome refers to the entirety of the epigenetic features possessed by an organism’s genome. DNA methylation was the first-recognized and is the most-studied epigenetic regulatory mechanism that plays a key role in the control of gene expression. In mammalian cells, DNA methylation mostly occurs at the cytosine of a CpG dinucleotide. CpG islands are genomic regions enriched in CpG dinucleotides. CpG islands cluster most at the 5’ promoter regions of many housekeeping genes and are generally hypomethylated to permit transcription, with the exception of genes involved in imprinting, X chromosome inactivation and cancer process [14,15,16]. The hyper-methylation of DNA combined with histone H3-lysine 9 [17] or histone H4-lysine 20 methylation [18] can result in inaccessible chromatin by recruiting chromatin remodeling proteins and generally causes the inhibition of downstream gene expression by physically blocking the binding of transcription factors [19]. Epigenetic modifications can be inherited from one cell generation to the next (mitotic inheritance) and between generations of a species (meiotic inheritance) and can be natural and essential for many developmental functions, but can also be influenced by several factors including age, environment, lifestyle or a disease state [20]. In mammalian cells, the changes in gene expression in response to external factors, such as viral infection, stress, drug delivery, temperature changes, dietary components, may have long-lasting effects on development, metabolism and health, sometimes even across generations. Epigenetics represent a link between the environment and gene expression [21].

In this study, we investigated the potential regulatory roles of epigenome signatures in the blood of dairy cows falling in the extreme category of the distribution of cortisol concentration in milk. We focused on DNA methylation in blood cells since blood is quickly accessible and highly informative of animal response to environmental challenges. Moreover, it was previously demonstrated that methylation levels at CpG sites in leukocytes could serve as an accurate predictor of CpG site variability in other tissues [22]. In parallel, we used transcriptome sequencing data to integrate the relationship between DNA methylation and transcriptional regulation on a genome-wide scale to identify novel genes involved in glucocorticoids secretion.

## 2. Material and Methods

### 2.1. Ethics Statement

The farms involved in the present study adhere to a high standard of veterinary care based on best practice manual, under the supervision of the official veterinary service. All experimental procedures and the care of animals complied to the Italian legislation on animal care (DL n.116, 27/1/1992) in force at the time the study was carried out and adhered to the bioethical rules of the University of Udine. The approval for conducting this study was also granted by the veterinarian responsible of animal welfare of the Department of Agricultural and Environmental Science of the University of Udine (Prot. N. 2/2015, OPBA—Organismo Preposto al Benessere degli Animali).

### 2.2. Animals Treatment and Experimental Design

The study was carried out ona commercial farm of the Italian Simmental breed located in the North East of Italy (Latitude: 45°76′491 N; Longitude: 13°10′466 E) in the month of May. From the herd, 126 lactating cows with days in milk (DIM) between 70 and 250, clinically healthy and with a parity from 2 to 6 (mean 3.0 ± 1.1) were initially selected for the study. One hundred ml of milk was collected from each cow at morning milking and an aliquot of 50 mL was transferred into a tube containing preservative and used for protein, fat, lactose analyses and for somatic cell count (SCC) determinations with FT-NIR (FOSS instrument, Hillerød, Denmark). The other aliquot of milk was transferred to a tube without preservative, frozen within 2 hand stored at −20 °C for cortisol analyses [23]. Milk cortisol was analyzed in skimmed milk, after centrifugation (1500 g, 4 °C, 15 min). Skim milk extracts were assayed by a solid-phase microtiter RIA [24], using a Microscint 20 instrument (Perkin-Elmer Life Sciences, Monza, Italy) and counted on the β-counter (Top-Count, Perkin-Elmer Life Sciences, Monza, Italy). All samples were assayed in duplicate. The sensitivity of the assay was defined as the dose of hormone at 90% binding (B/B0) and was 3.125 pg/well. The intra-assay and inter-assay coefficients of variation in high and low cortisol pooled skim milk samples were 5.9% and 9.1% and 13.5% and 15.1%, respectively. On the same day, after milking and before feeding, blood was sampled form the tail vein in a tube with K_3_-EDTA (Venoject, Terumo Europe N.V., Leuven, Belgium) and in one PAXgene Blood RNA System tube (Preanalytix, Hombrechtikon, Switzerland) for analysis. The K3-EDTA tubes were centrifuged within 1 h at 755× *g* for 10 min at 20 °C and the blood cells stored at −20 °C for DNA analysis. The PAX gene tubes were handled and stored following the manufacturer’s instructions for RNA extraction. The body condition score (BCS) of each cow was also recorded using a scale from 1 (thin) to 5 (fat) with 0.25-point intervals [25] (Table 1). Cows were ranked on the basis of cortisol in their milk and the 10 cows with the highest and the 10 cows with the lowest concentrations were selected for transcriptome and epigenome analysis (Figure 1).

### 2.3. DNA Isolation, Library Preparation and RRBs Sequencing

Blood samples from these 20 animals were collected. Peripheral blood mononuclear cells (PBMC) were isolated from heparinized blood samples with Ficoll-Paque. CD4+ lymphocyte cells were isolated with the CD4 anti-bovine monoclonal antibody (VMRD, Pullman, WA, USA) and positively selected with GAM microspheres using the MACS MiltenyiBiotec system (BergischGladbach, Germany). DNA was extracted from CD4+ cells following the QIAamp DNA Blood Midi Kit (Qiagen) procedures. gDNA concentration was measured with Quant-it Picogreen dsDNA assay and 1 μg input was used in the MSP1 digest. Following overnight incubation at 37 °C, digestion reactions were terminated by adding 0.5 M EDTA and purified on a GeneJET PCR purification column. Libraries were prepared using the NEB Next Ultra DNA library preparation kit for Illumina and methylated adapters. Subsequently, ligated products corresponding to DNA fragments 150–400-bp long were isolated and purified using 2.5% agarose gel electrophoresis. The recovered DNA was bisulfite converted using the EZ DNA Methylation Gold kit (Zymo Research). DNA with a known methylation level was used as a spike control, and all conversion rates were >99%, ranging from 99.01 to 99.39% (Appendix A). Fourteen cycles of PCR were performed, and the products were purified using AMPureXP beads. Reduced Representation Bisulfite Sequencing (RRBS) libraries were used for cluster generation and subsequent sequencing on Illumina HiSeq2500 PE 2 × 50 bp (NXT-Dx Ghent, Belgium, http://www.nxt-dx.com/).

### 2.4. Data Analysis

Preliminary quality control of the raw reads was carried out with FASTQC v0.11.9 (http://www.bioinformatics.babraham.ac.uk/projects/fastqc/). The FASTQ sequence reads were generated using the Illumina Casava pipeline 1.8.2. The quality and adapter trimming of Illumina raw sequences was performed with Trim Galore v0.6.1 (http://www.bioinformatics.babraham.ac.uk/projects/trim_galore/) using a two-step approach, which allowed us to remove two additional bases containing a cytosine, which were artificially introduced in the end-repair step during the library preparation. Bismark software (version 0.22.1) [26] was used to align each bisulfite-treated read to the bovine reference genome (ARS-UCD1.2) with option-N 1 (maximum number of mismatches allowed). The reference genome was first transformed into a bisulfite-converted version (C-to-T and G-to-A conversions) and then indexed using bowtie2 software [27]. Sequence reads were also transformed into fully bisulfite-converted versions (C-to-T and G-to-A conversions) before they were aligned to similarly converted versions of the genome in a directional manner. Sequence reads that produced the best unique alignment from the two alignment processes (original top and bottom strand) were then compared to the normal genomic sequence, and the methylation state of all cytosine positions in the reads was inferred using the *Bismark_methylation_extractor* function. Read duplicates were marked and removed using Picard Tools v2.2 (http://broadinstitute.github.io/picard).

### 2.5. Identification of DMRs and DMGs

Differentially methylated regions (DMRs) were identified within the statistical environment R using the library methylKit [28], which applies a sliding-window approach. The window was set to 1000 bp and the step size to 500 bp. In order to ensure the good quality of the data and great confidence in the methylation percentage, we first filtered out bases with less than 10 reads or more than the 99.9th percentile of coverage distribution. Coverage values were normalized by default and bases were merged in order to retain the ones that were covered in all samples. A logistic regressionwas then implemented to calculate *p* values, that were adjusted to *q* values using the Sliding Linear Model (SLIM) method [29]. We excluded covariates and overdispersion correction from the model since they showed not to provide significant changes to the results. DMRs regions were defined those that had *q* value of less than 0.05 and a read coverage of greater than ten in all samples. When a region where a DMR and a specific gene function element overlapped, the corresponding gene was selected as the DMR-related gene, namely a differentially methylated gene (DMG). Gene features within differential methylated genes (DMGs) were annotated by matching information available in the General Feature Format (GFF) file downloaded from the Ensembl database, release 99 (ftp://ftp.ensembl.org/pub/release-99/gff3/bos_taurus/Bos_taurus.ARS-UCD1.2.99.gff3.gz). The promoter region was defined as an area 2-kb upstream of the transcription start sites.

### 2.6. Association Analysis

Correlations between DNA methylation levels and gene density, chromosome length and GC percentage were measured using Pearson’s product–moment correlation coefficient. Correlations between DNA methylation and gene expression were measured using Spearman’s rank correlation coefficient because the relationship of DNA methylation with gene expression data was not necessarily expected to be linear. All these analyses were performed with custom R scripts.

### 2.7. GO and KEGG Enrichment Analysis of DMR-Related Genes

Gene Ontology (GO) and KEGG pathway enrichment analysis of DMGs were performed by *clusterProfiler*, an ontology-based R package able to automate the process of biological term classification and enrichment analysis of gene clusters and to provide a visualization module for displaying analysis results [30]. GO terms and KEGG with *p* values of less than 0.01 and *q* values of less than 0.2 were considered significantly enriched by DMR-related genes. The DMGs involved in the GO pathways related to development and metabolism were analyzed with the R package *enrichplot* [31] to highlight links between genes and GO terms in a gene concept network.

### 2.8. Sample Preparation for RNA—Seq Analysis

Blood samples collected in the PAX gene Blood RNA Tubes (PreAnalytiX GmbH, Hombrechtikon, Switzerland) were frozen 4 h after the collection and stored at −80 °C until the RNA isolation. Prior to RNA isolation, blood samples were thawed at +4 °C for at least 12 h. RNA was isolated according to the PAX gene Blood RNA Kit (PreAnalytiX GmbH, Hombrechtikon, Switzerland) protocol. The quantity and quality of RNA were analyzed with Agilent 2100 Bioanalyzer (Agilent Technologies, Santa Clara, CA, USA) before sequencing and the RNA integrity number (RIN) score was ≥7 for all the samples. Since the RNA-Seq is highly reproducible within a large dynamic range of detection and provides an accurate estimation of RNA concentration in peripheral whole blood [32]; globin depletion was avoided, as it could significantly reduce the amount and quality of isolated RNA. Sequencing was performed with the Illumina pipeline.

### 2.9. Quantification of Gene Expression Levels and Differential Expression Analysis

After quality control and the filtering of raw data, clean reads were aligned to the bovine reference genome (ARS-UCD1.2) using STAR v2.7.3 [33], a splice-aware aligner. featureCounts (Subread package v1.6.5) [34] was used to count the read numbers mapped to each gene. Differential gene expression between the high and low groups was performed using the DESeq2 R package (1.26.0) [35]. The *p* values were adjusted using the Benjamini–Hochberg method [36]. A corrected *p* value of 0.05 was set as the threshold for significantly different expression, while genes with extremely low expression (fold changes (FC) values between +1 and −1) were filtered out.

## 3. Results

### 3.1. Global Mapping of DNA Methylation

In the present study, blood samples from 20 Italian Simmental dairy cows falling in the high and low tails of the distribution of milk cortisol concentration (Figure 1) were used to investigate genome-wide DNA methylation. A total of 278 and 262 million raw reads were generated from the bisulfite sequencing of the high and low groups, respectively. After data filtering, 139 and 132 million clean reads were generated; of these, approximately 48 and 46 million mapped to the reference genome with a unique best fit. The mapping efficiency of high- and low-variant samples ranged from 25.6% to 41.5% and from 30.9% to 42.8% of the cattle genome, respectively (Table 2 and Appendix A).

### 3.2. DNA Methylation Patterns

In the genome of each group, over 50% of the CpG sites were methylated, which is the primary DNA sequence context of cytosine methylation. Specifically, we observed, on average, genome-wide levels of 50.5% CG, 6.8% CHG, and 7.8% CHH (where H is A, C, or T) methylation in the high-variant group and 53.6% CG,7.1% CHG, and 8.1% CHH methylation in the low-variant group. (Table 3). Compared with the low group, among the methylated cytosines the relative proportion of CG in the high group was greater (73.24% vs. 68.84%) and consequently that the proportion of methylated CHH and CHG was lower (17.94% vs. 21.16% and 8.83% vs. 10% respectively; Figure 2).

We noted that a high cortisol concentration in milk resulted in a decrease in overall methylation at all sites in comparison to low cortisol concentrations, but in an increase in the methylation rate of CpG sites compared to non-CpG sites, especially in the CHH context.

### 3.3. DNA Methylation Levels of Gene Features

To disentangle the differences identified in global DNA methylation profiles across high- and low-cortisol dairy cows, we compared the average DNA methylation levels of different gene features along the genome (Figure 3).

A major proportion of CG methylated sites were present in the regions of 3’UTR followed by intron regions, while the average methylation levels of promoters,5’UTR and exon regions were the lowest. Conversely, all five regions exhibited a more balanced distribution of methylation at CHG and CHH sites.

### 3.4. Differential Methylated Regions (DMR) and Genes (DMG)

A total of 897 DMRs were identified between the two groups (*q* value ≤ 0.05), falling into 248 DMGs (Appendix A). Of these DMGs, 146 were up-methylated, and 102 genes were down-methylated in the high group. Moreover, we also detected a negative correlation of methylation levels with gene density (Pearson’s *r* = −0.712, *p* value < 0.001) and with chromosome length (Pearson’s *r* = −0.754, *p* value < 0.001) and a positive correlation with the GC percentage (Pearson’s *r* = 0.958, *p* value < 0.001). The differentially methylated regions were mainly located in introns, whose proportion covers nearly half of the total percentage of methylation, followed by the exons and the promoter regions (Figure 4).

### 3.5. Functional Enrichment Analysis of the DMGs

To investigate the potential biological functions of the DMGs, a GO enrichment analysis and a KEGG pathway analysis were performed. All DMGs were annotated in three GO categories: biological process; cellular component; and molecular function. Some of these DMGs were enriched in the following biological process terms: cellular process (129; 52%); single-organism process (123; 49.6%); and metabolic process (103; 41.5%). In addition, most of the top 10 significantly enriched pathways in GO analysis were strictly related to nervous system activity and metabolic responses to stress (Figure 5 and Appendix A).

Several DMGs, such as *IGF2, LIF, NR5A1, SRF, FOSB, PDE5A, TSC1, NRN1, WNT6* and *HOXB1*, were involved in biological processes significant for stress response, inflammatory reactions and the immune system. Furthermore, the gene–gene interaction network analysis showed that these DMGs were highly correlated with each other (Figure 6).

The KEGG enrichment analysis identified 62 pathways (top 20 in Figure 7 and Appendix A).

Of these pathways, some were associated with immune system and glucocorticoid secretion, such as the T cell receptor signaling pathway (*q* value = 0.051), the hedgehog signaling pathway (*q* value = 0.032), axon guidance (*q* value = 0.041), the calcium signaling pathway (*q* value = 0.157) and the GABAergic synapse pathway (*q* value = 0.188). Eleven differentially methylated genes participated in these five pathways.

### 3.6. Differentially Expressed Genes (DEGs)

A total of 324 DEGs were also identified in the present study between the two groups (*q* value ≤ 0.05). Among these, 149 genes were over-expressed and 175 were under-expressed in the high group (Appendix A). We mapped these genes to the GO and KEGG pathway. Again, they were annotated in three GO categories: biological process; cellular component; and molecular function. Some of these DEGs were enriched in the following biological process terms: cellular process (104; 36.1%); single-organism process (95; 33%); and metabolic process (75; 26%). For all the DEGs, only the GO term “binding”, with 58 genes in the categories of molecular function, was significantly enriched (*q* value < 0.05), implying that a broad range of genes experienced transcriptional regulation during the activation process of cortisol secretion. A KEGG pathway analysis was also performed to investigate pathways in which DEGs might be involved. In the top 20 list (Appendix A), pathways implicated in inflammatory reactions (Inflammatory bowel disease (IBD), *q* value= 0.014), and immune defense functions against viral disease (Influenza A, *q* value = 0.011, Hepatitis B, *q* value = 0.109, Hepatitis C, *q* value = 0.103) are found. This confirms several studies that recently indicated a strong correlation between stressors and the progression and outcome of liver inflammatory diseases, such as chronic viral hepatitis [37]. To examine the relationship between DNA methylation and genome-wide gene expression, an association analysis was also performed between these DMGs and differentially expressed genes (DEGs) obtained from transcriptome data from the same animals (*p* value < 0.001). We found six overlapping genes, four were up-methylated, *TRIM26, PAX2, SYNGR1* and *SNCB*, and two down-methylated, *UPP1* and *HTRA1* (Appendix A and Table 4).

## 4. Discussion

There is a growing debate in the livestock industry about stress and its detrimental effects on animal welfare and psychological and physiological responses of animals to traditional agriculture procedures and new production technologies [38]. The measurement of corticosteroid hormones is commonly used as an indicator of the animal’s response to stress. Blood sampling is a stressful factor, so, in recent years, several efforts have beenmade in many species to find alternative and non-invasive ways to measure cortisol level [39,40,41]. Moreover, the cortisol concentration in the blood of cattle can vary due to circadian rhythmicity and several extrinsic factors, such as cold, heat, humidity and wind [42]. On the contrary, milk sampling can be achieved directly in a milking parlor without animal handling and overcomes some of the problems associated with other sampling sites, such as blood, urine and feces. Furthermore, several studies showed that free cortisol concentrations in milk aredirectly related to free cortisol in the blood in cows, especially after adrenal stimulation [43,44]. These works suggest that milk cortisol concentration can be considered a valid proxy of blood cortisol.The use of milk cortisol as a biomarker of environmental stimuli in dairy cows was also reported by Tsukada et al. (2008) [45], Waki et al. (1987) [46], Verkerk et al. (1998) [47] and Poscic et al. (2018) [20]. For this reason, milk can be considered a preferential site of sampling in dairy cows to point out the short-term stimulation of the HPA axis [48].

Blood is a peripheral tissue and is not necessarily related to changes in DNA methylation in the central nervous system. Despite this, we focused on DNA methylation in blood cells since several studies recently proved that peripheral tissues methylation can be informative for the neurobiological mechanisms underlying high cortisol levels. First, cortisol is released into the periphery by the pituitary and is known to affect multiple tissue types [49]. Second, HPA axis genes are highly expressed in peripheral blood mononuclear cells [50]. Third, it was previously demonstrated that methylation levels at CpG sites in leukocytes could serve as an accurate predictor of CpG site variability in other tissues, especially the brain [22,51]. Peripheral changes in methylation may therefore at least partially be considered as proxies of epigenetic processes in the brain and provide highly informative evidence of animal responses to environmental challenges.

To the best of our knowledge, this work is the first attempt to investigate the underlying epigenetic mechanism related to the expression of genes triggered by glucocorticoid secretion and compare the genome-wide methylation profiles between two groups of dairy cattle with high and low levels of milk cortisol. Usually, stressful factors lead to the adrenal secretion of glucocorticoids initiated by the hypothalamic neuropeptide corticotropin-releasing hormone (CRH), which stimulates pituitary release of the adrenocorticotropic hormone (ACTH). The activation of ACTH receptors in the adrenal cortex stimulate glucocorticoid synthesis and secretion; glucocorticoids then act on a wide range of target tissues [52]. Most of the diseases induced by an impaired stress response arise from primary defects in the adrenal glands that are usually associated with significantly decreased neutrophils and natural killers (NKs) [53].

It is well known that DNA methylation, especially in the promoter regions, can change gene expression via different modes [54]. Our results indicated that only a small proportion of the DMRs were located in the 5′UTR, 3′UTR, and promoter regions of the cattle genome, while nearly half of the total percentage of methylation falls within intron regions (Figure 4).

### 4.1. Key Differentially Methylated Genes Associated with Cellular Defense and Stress Response

In the present study, we identified several DMGs associated with stress response as assessed by cortisol level. Some of these, including *PDE5A, IGF2, NR5A1, FOSB, NRN1* and *WNT6* appear to be of particular interest because of their functions with cattle immune system. Furthermore, these DMGs, which are already known to be involved in adaptive responses to numerous stressors from human studies, could participate to the same molecular mechanisms in cattle [55,56]. Genes that act as regulators in immune mechanisms are strong candidates for differential methylation as DNA is isolated from white blood cells. Heat stress is certainly one of the most problematic kinds of stress able to decrease the welfare and productive performance of dairy and beef cattle [57]. *PDE5A*, which encodes for cGMP-binding, a cGMP-specific phosphodiesterase, besides having one of the top hypo-methylated intronic regions, as shown in our experiment, has also been identified as hypo-methylated in the liver and mammary gland tissues of bull calves and heifers exposed to heat stress during pregnancy, again with the hypo-methylated window mainly located in an intronic region, as reported by Skibiel et al. (2018) [58]. This gene is also known to be part of different pathways, biological functions, and molecular processes such as cell signaling, protein binding, phosphorylation, cell activation and cGMP binding [58]. Insulin-Like Growth Factor 2 (*IGF2*) belongs to the IGF signaling pathway, a highly conserved evolutionarily network that regulates cell proliferation, differentiation, survival and longevity [59,60,61]. Interestingly, there is evidence that this gene increases the expression of interleukin (IL)-10 from specific B cells and plays a crucial role in inhibiting intestinal allergic inflammation, as demonstrated by Geng XR et al. (2014) [62] in an experiment with a mouse model. It has also previously been proven that *IGF2* is one of the most complex and well-characterized imprinted genes, both in mice and humans [63]. A paternally expressed quantitative trait loci (eQTL) affecting muscle growth, fat deposition and the size of the heart in pigs maps to *IGF2* and is caused by a nucleotide substitution in intron 3 of this gene [64]. Normally, *IGF2* is highly expressed during embryo development and then its expression drops at weaning and becomes undetectable, but Barroca et al. (2017) [65] also recently observed a significant gene activity in mice during adulthood for maintaining tissue homeostasis. They also noted that reducing or slowing down the *IGF2* level would represent a means for stem cells to survive when faced with cellular stressors, resulting in increased cellular health. Vangeel et al. (2015) [66] demonstrated in humans that maternal emotional stress during pregnancy, as defined by cortisol measurements, is associated with fetal DNA methylation of *IGF2*.

Nuclear receptor subfamily 5 group A member 1 (*NR5A1*) encodes for steroidogenic transcription factor 1 (SF-1), a key regulator of adrenal function and reproductive development. It was suggested that a normal dosage of SF-1is required for mounting an adequate stress response in mice since a reduced expression of this transcription factor would lead to adrenal failure [67]. Another interesting transcription factor that has been found to be differentially methylated in our gene list is *FOSB*. This gene encodes for a protein implicated as a regulator of cell proliferation, differentiation, and transformation. Previous studies carried out in human shave shed light on its role as regulator in stress response, since it has been observed that, during repeated stress events, ΔFosB, an alternative splice product of the *FOSB* gene, accumulates in several brain areas and start to induce a reduction in the deleterious effects of chronic stress, such as depression-like behaviors and despair [68,69]. Interestingly, this transcription factor activity seems to be strictly linked to *CREB*, another stimulus-induced transcription factor, which acts together with *FOSB* in histone acetylation and deacetylation, mediating long-lasting forms of synaptic plasticity [70]. Furthermore, the *CREB* gene has been previously identified as a differentially methylated gene in pigs during a heat stress study by Hao et al. (2016) [71]. This evidence suggests that *FOSB* and *CREB* could act together in an epigenetic mechanism involved in stress response.

We found two other interesting genes which exhibited hypomethylation following an increase in cortisol level. They are also involved in all the top GO significant pathways linked to nervous system development and neurogenesis. The first is *NRN1*, which encodes a member of the neuritin family, an extracellular glycophosphatidylinositol-linked protein that promotes neuronal survival, differentiation, function, and repair, even if the exact mechanism of this neuroprotective effect remains unclear [72]. Recent studies demonstrated that the atrophy of neuronal processes contributes to the negative effects of stress, but also that this process is reversible. Hyeon Son et al. (2012) [73] suggested a connection between neuritin and the prevention of stress effects by protecting the brain from the atrophy of dendrites and spines. The authors used a viral vector to increment neuritin expression in the hippocampus of some rats, and those with an increase in neuritin mRNA levels did not show adecrease in sucrose preference resulting from chronic stress and also did not display the reduction in dendritic spine density that appears with chronic stress either. Besides being crucial for the development, survival and function of neurons, this gene also promotes the maintenance of regulatory T cells (Tregs), a subpopulation of T cells that modulate the immune system, maintain tolerance to self-antigens, and prevent autoimmune disease [74]. Barbi et al. (2016) [75] proved that a knockout of the neuritin gene in Tregs could lead to the onset of inflammatory/autoimmune diseases. The second gene that belongs to GO nervous system pathways is *WNT6*. It belongs to gene family of structurally related genes that encode secreted signaling proteins. These proteins have been implicated in oncogenesis and in several developmental processes, including the regulation of cell fate and patterning during embryogenesis [76]. Most functional studies indicate that the Wnt family, including *WNT6*, exerts pro-inflammatory functions on different cellular targets, including various types of immune and non-immune cells [77]. This is consistent with recent studies about glucocorticoids, since they can influence a broad range of both innate and acquired immune responses, while a variety of regulatory proteins may also mediate the anti-inflammatory effects of glucocorticoids [78]. Apart from IGF2, all of the above mentioned genes exhibited hypo-methylation within intronic regions. We also found that homeobox genes were significantly overrepresented in the differentially methylated gene list. This is not surprising, since the methylated activation or deactivation of these transcription regulators starts a cascade reaction that targets different pathways, thus allowing the fast response of the organism to the stressful situations. Furthermore, homeobox genes are reported to be differentially expressed in leukocytes [79]. The homeobox genes are clearly better studied in embryonic development where they orchestrate the body plan [80]. Recent evidence, however, points out the essential regulation role of the homeobox genes in adulthood, both in normal and pathological physiological processes. We will now provide a brief description of each of the differentially methylated homeobox genes, focusing on their biological role in adult vertebrate, where known. Ceramide synthase 4 (*CERS4*) synthesizes ceramides containing C18-22 fatty acids. Thanks to this participation in the sphingo lipid metabolism, it likely plays a role in the control of body weight and food intake [81]. The function of the homeodomain in this protein is unknown but its deletion in CerS5 does not affect activity [82]. The cut-like homeobox 1 protein (cux1) is a transcription factor that regulates a large number of genes and microRNAs involved in multiple cellular processes [83]. *CUX1* encodes two main isoforms with p200, which binds to DNA with extremely fast kinetics (rapid “on” and “off” rates), while, usually, the classical transcription factor binds stably to DNA [84]. The p100 *CUX1* isoform shows normal slow DNA binding kinetics and it functions as a transcriptional repressor or activator [85]. Distal-less homeobox 3 gene (*DLX3*) is a transcriptional activator that regulates keratinocyte proliferation and differentiation, with the assistance of the tumor suppressors p53 [86] and p63 [87]. A better knowledge of the biological mechanism of keratinocyte growth and differentiation could produce important returns for bovine production [88,89]. The transcriptional role of homeobox B1 (*HOXB1*) during embryonic development involves the anterior bodily structures. A relatively high level of expression has been registered in the brain stem of the adult brain, also in territories where *HOX* genes are detected during development. For example, the developmental expression of *HOXB1* is linked to a subpopulation of noradrenergic neurons, a collection of neurons located in the central nervous system. Once activated, they are able to decrease anxiety-like behavior and induce an active coping strategy in response to acute stressors [90]. The iroquois homeobox 6 gene (*IRX6*) is involved not only in lactation [91], but also in neuronal development [92]. The LIM homeobox 5 gene (*LHX5*) is, again, involved in brain development, both of the mammillary body [93] and the forebrain [94]. Its methylation level has been found altered in a mouse model of neuro-developmental disorders [95].

### 4.2. Key Differentially Methylated Genes Associated with KEGG Pathways

The regulation of the stress response system is a complex biological process involving not only different functional areas of the brain like the amygdala, hypothalamus and prefrontal cortex, but also different tissues like the adipose tissue, bones, liver, muscles and pancreas. Therefore, examining regulatory networks is the preferred method of analysis. In the present study, the DMGs were enriched in several KEGG pathways, including the calcium signaling pathway, T cell receptor signaling pathway, axon guidance and GABAergic synapse.

Glucocorticoids can influence a broad range of both innate and acquired immune responses and it is well known that they not only have anti-inflammatory effects, but also induce pro-inflammatory responses [96]. Among the significant pathways, the T cell receptor signaling pathway has been studied in cattle for its role in stress defense or stress-related diseases. For example, a variation in T-lymphocyte levels following heat stress in two *Bos taurus* and *Bos indicus* crossbreeds was previously reported [97]. Interestingly, we found, within this pathway, two DMGs, *PAK4* and *LCK*, related to the nervous system both in normal and pathological physiological processes [53,56]. *PAK4* regulates a wide range of cellular functions, it is essential for embryonic brain development and has a neuroprotective function. Lately, the transcription factor *CREB* has emerged as a novel effector of *PAK4* [98]. As mentioned before, this gene seems to also be involved with *FOSB* in a mechanism that modulates synaptic plasticity. This finding has broad implications for the role of *PAK4* in health and disease and suggests the presence of a complex interaction between several genes that could play a role in stress response. Several studies demonstrated that GABAergic synapse and axon guidance pathway activity undergo significant changes in response to acute stress, which remodel the excitability of neurons and the activation of the HPA axis [99,100,101]. In addition, increases in dexamethasone corticosteroids are also associated with a decline in the Ca^2+^concentration within the endoplasmic reticulum lumen, likely an effect of increased aldosterone secretion, which contributes to the imbalance of total cellular Ca^2+^ [102]. All this confirms previous studies showing that changes in methylation obtained from peripheral blood mononuclear cells were significantly enriched for central nervous system pathways [103]. Additional studies of the translational and posttranslational effects together with the expression and function of the proteins encoded by the genes identified here will be required to provide a global view of the methylation mechanisms undergoing variations in cortisol secretion.

## 5. Conclusions

In summary, this is the first study to compare comprehensive DNA methylation profiles as well as transcriptome data of dairy cattle populations in relation to cortisol secretion. Further studies are also needed to explore the function of non-CpG methylation, which might help to improve our knowledge about the biological significance of the non-CpG methylation changes that occur during stress processes.

We identified DMRs and genes associated with these regions. Pathway and network analyses of these differentially methylated genes revealed a number of candidate genes that might affect different cortisol levels or at least could be activated as a consequence of glucocorticoid synthesis and secretion. The results of this study might therefore provide additional insight into the epigenetic genome mechanisms related to stress response and will likely contribute to the improvement of animal welfare.

## Figures and Tables

**Figure 1 genes-11-00850-f001:**
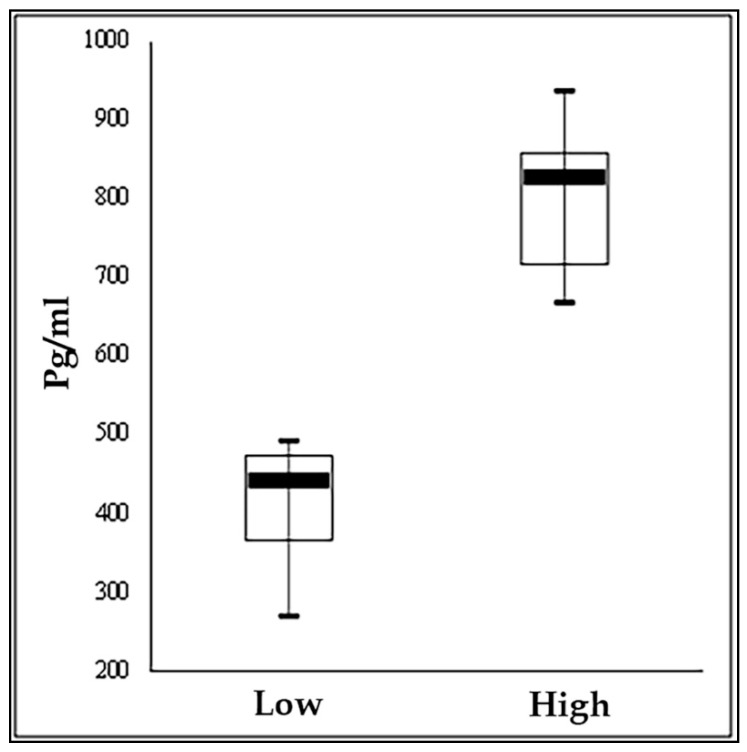
Box plot of milk cortisol distribution of 10 cows with the higher and the 10 cows with the lower values.

**Figure 2 genes-11-00850-f002:**
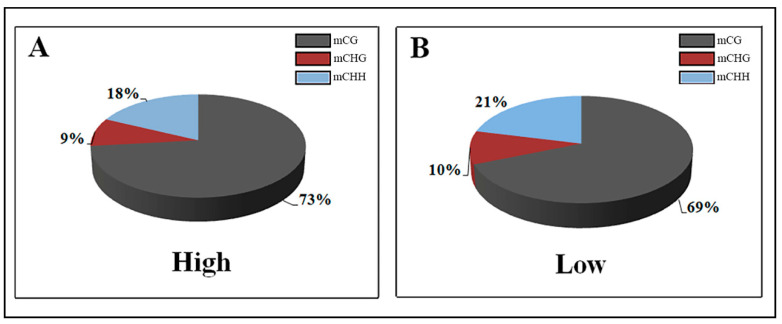
Comparison of DNA methylation patterns in the two high-and low-cortisol groups.

**Figure 3 genes-11-00850-f003:**
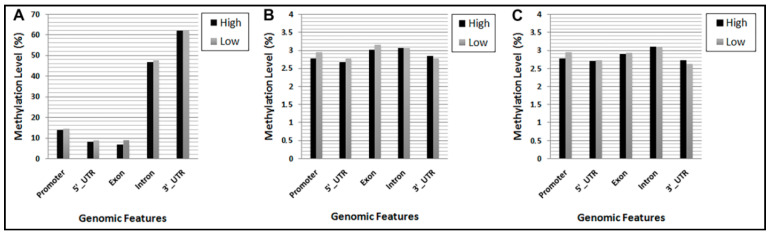
DNA methylation levels of different functional regions between the two high- and low-cortisol groups. (**A**) CG regions. (**B**) CHG regions. (**C**) CHH regions. H = A, C, or T.

**Figure 4 genes-11-00850-f004:**
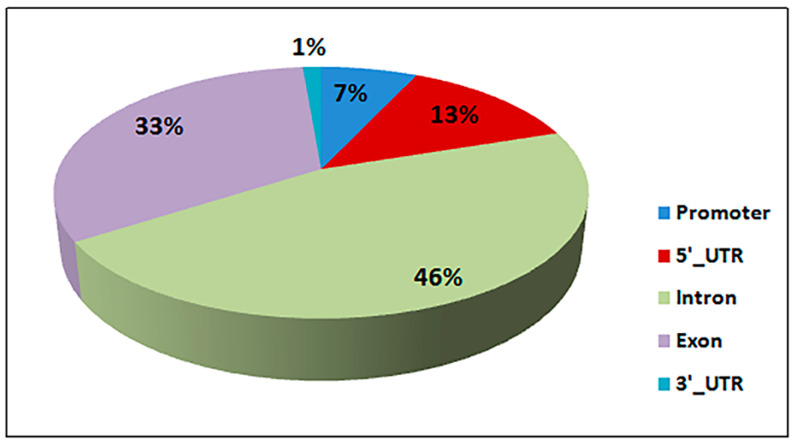
Distribution of differentially methylated regions (DMRs).

**Figure 5 genes-11-00850-f005:**
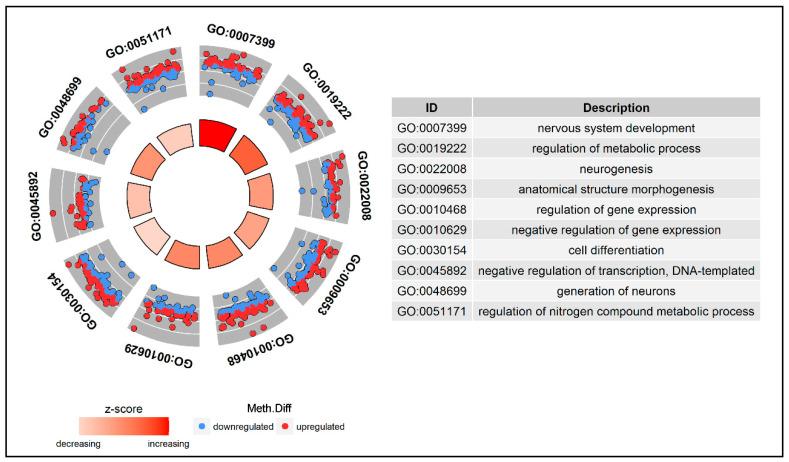
Gene Ontology (GO) circle plot for differentially methylated genes (DMGs). The inner ring is a bar plot where the height of the bar indicates the significance of the term (*q* value), and color corresponds to the z-score. The outer wheel shows a scatter plot of methylation difference for each gene under the Gene Ontology(GO) terms. Red dots indicate hyper-methylated genes and blue dots show hypo-methylated genes.

**Figure 6 genes-11-00850-f006:**
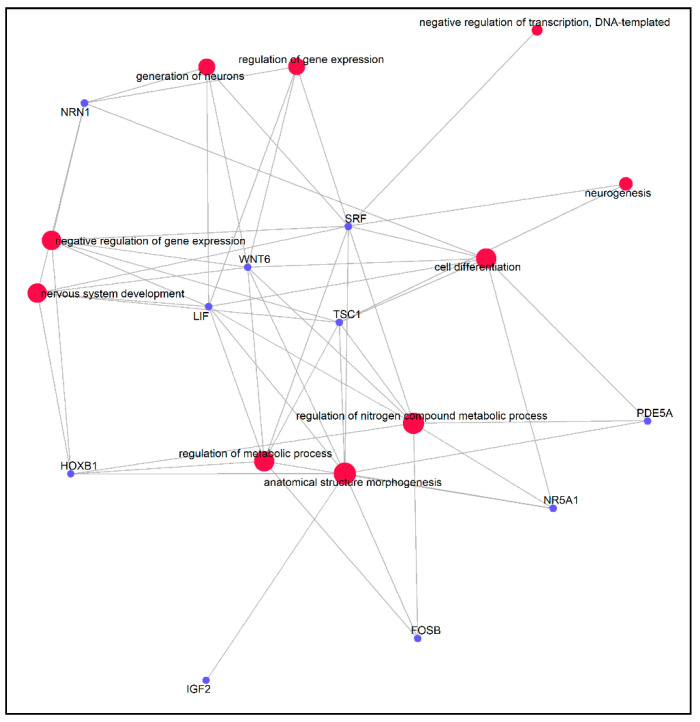
Gene-pathway association network indicating genes affiliated to significantly GO enriched pathways. Red dots indicate GO pathways and blue dots indicate genes.

**Figure 7 genes-11-00850-f007:**
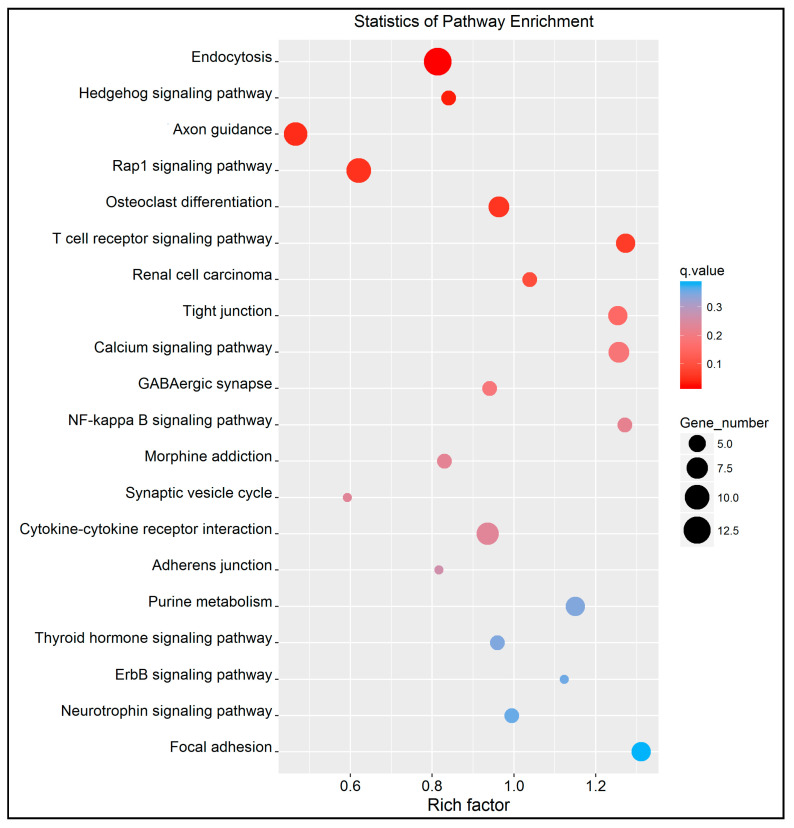
Scatter plot of KEGG pathway enrichment statistics. Top 20 statistics of pathways, enrichment in the KEGG cattle database. The *y*-axis represents the name of pathway and the *x*-axis represents the rich factor, the proportion of differentially methylated genes to all the genes that are annotated in a specific pathway term. Dot size represents the number of genes and the color indicates the *q* value.

**Table 1 genes-11-00850-t001:** Body condition score (BCS), parity, days in milking (DIM) and milk yield and quality in the 10 cows with lowest and 10 cows with highest cortisol concentration in milk.

Item		Low Cortisol		High Cortisol		
		Mean	Sd	Mean	Sd	*p* Value
BCS	Score	3.23	0.38	3.10	0.47	0.262
Parity	*n*°	1.70	0.95	2.10	0.88	0.170
DIM	Days	156.2	62.8	143.3	69.3	0.334
Milk	Kg/d	26.20	6.54	30.38	6.23	0.080
Fat	%	3.68	1.03	3.95	0.70	0.254
Protein	%	3.61	0.25	3.59	0.39	0.441
Casein	%	2.84	0.20	2.81	0.32	0.415
Urea	mmol/L	21.39	5.12	21.01	6.13	0.442
SCC	Number	133.50	89.09	153.40	110.02	0.331
Cortisol	pg/mL	399.9	79.7	814.8	88.6	0.000

Before ranking, cows with somatic cell counts (SCC) higher than 200,000 cells/mL were excluded.

**Table 2 genes-11-00850-t002:** Data generated by genome-wide bisulfite sequencing. high- and low-variant tails of the distribution of milk cortisolconcentration.

Samples	Raw Reads	Clean Reads	Mapped Paired End Reads	Average Mapping Rate (%)
High	278,881,124	139,639,195	36,779,144	35.15
Low	262,917,162	132,458,581	35,048,383	35.71

**Table 3 genes-11-00850-t003:** Genome-wide methylation levels of the two high- and low-cortisol groups for CpG and non-CpG sites.

Samples	mCpG(%)	mCHG(%)	mCHH (%)
High	50.48	6.82	7.82
Low	53.58	7.11	8.14

**Table 4 genes-11-00850-t004:** Six DMGs that overlapped with differentially expressed genes (DEGs) (*q* value < 0.05 in both analyses).

Gene ID	Gene Name	DMRs	Methylation Stat (High vs. Low)	UP/DOWN Regulation (High vs. Low)
ENSBTAG00000035744	*TRIM26*	exon	Hyper	Down
ENSBTAG00000021566	*PAX2*	exon, intron	Hyper	Up
ENSBTAG00000005765	*SYNGR1*	intron	Hyper	Up
ENSBTAG00000009803	*SNCB*	utr3, exon, promoter	Hyper	Down
ENSBTAG00000008428	*UPP1*	intron	Hypo	Down
ENSBTAG00000008389	*HTRA1*	exon	Hypo	Up

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
