# Peer review of "Genome-Wide DNA Methylation and Gene Expression Profiles in Cows Subjected to Different Stress Level as Assessed by Cortisol in Milk"

_genes, 2020, doi:10.3390/genes11080850_

Round 1

Reviewer 1 Report

In the study reported by Del Corvo et al., an interesting and original approach is performed. The aim of the study is to analyze the methylome and transcriptome in blood cells in relation with the stress status of cows. This approach could have great interest in dairy cows to develop biomarkers of the stress and to adapt the management of animals.

The experimental design is correct with a good definition of the two experimental groups (High and low milk cortisol, analyzed as stress marker). It is also adequate in term of number of cows per group (n=10 per group). The inclusion criteria of individuals were correctly described and justified. The methodologies, RRBS for DNA methylation analysis and RNASeq for gene expression are in accord with the aim of this study.

Nevertheless, the identification of differentially methylated features is strongly compromised by major methodological issues. :

The description of RRBS methodology is very laconic; important information and quality controls to validate the methodology have been omitted.

- The first step of RRBS is an enzymatic cleavage. MspI is frequently used but it is not the generality. Please indicate the enzyme used, since the size of generated fragments and the genomic features targeted by RRBS are dependent of the enzyme.

- The size selection of the fragments (150-175bp and 175-225 bp) is not classical and could be justified (in general in bovine, fragments from 150 to 400 bp including the adapters are selected; Zhou et al., 2016; Perrier et al., 2018). The methodology to size-select the fragments was omitted.

- Only one run of DNA conversion is performed. The efficiency of bisulfite conversion is not mentioned. The authors reported the presence of methylated cytosines in CHH and CHG contexts (mCHG 18-21%; mCHG 9-10%). These percentages are very high and never described when a good efficiency of bisulfite conversion is obtained (more than 99%; see as previously, Zhou et al., and Perrier et al.).

- A total of 278 and 262 million raw reads were generated. These data concern all the ten libraries per group: please, give the overview data for each library.

The mapping efficiency varies to 25.6% to 42.8%: these strong variations of mapping are not understandable.

- The authors use the statistical environment R using methylkit with a sliding – window approach, but the methylation difference threshold between the two groups to identify the DMR is not indicated. In supplementary table 1, the difference of methylation varies from 17% to 20% for 300 DMRs targeting a gene. Nevertheless, a large majority of DMR displays a lower methylation difference (288 DMRs with methylation difference between -10% and +10%. The authors should rise the methylation difference threshold to at least 15%. What is the coverage threshold used to retain DMRs in the differential analysis? This should be clearly stated in the Method section.

The low raw reads per library, the trouble of bisulfite conversion, the lack of coverage information to define DMR and the lack of methylation difference threshold represent major problems could create doubt about the validity of these data. Moreover, technical validation of several DMRs using an independent technology is mandatory given the above remarks.

Results

Fisrtly, the efficiency of bisulfite conversion should be verified (see Chatterjee et al.). I think that new libraries should be performed.

After definition of the threshold of methylation difference, please, indicate first the genomic localization of all DMRs (distribution between inter and intra genic regions) and then select the DMRs targeting genes. Please indicate how the promoter regions have been defined taken account the bovine genome annotation. The sentence line 205-208 should be certainly rewritten.

- Please explain how the negative correlation is found between methylation level and gene density? (line 212).

In functional enrichment analysis, have the authors taken in consideration the genome part analyzed by RRBS (only 3% in bovine)?

The authors presented also transcriptomic data. I am not specialist of RNA seq analysis. A heat map of DEG could be presented. These data appear unmistakable. Surprisingly, they are few used. For example, from the DEG six genes overlapp with the list of genes targeted by a DMR. The authors should use these six genes/DMRs to validate the RRBS data.

Discussion

The DEG data and DEG/DMR data should be more discussed.

Reviewer 2 Report

The manuscipt addresses interesting task, and applied methods are appropriate and timely, results clearly presented.DNA methylation pattern was slightly dfferent in the high and low cortisol group of Italian Simmenthal cattles. Fig 3 the A,B and C labells are missing. The study identified differential methylated regions and genes. The former was predominant in the introns, this  interesting result just mentiond in chapter 4. but not discussed in the light of other publications if any. On the contrary the differentially methylated genes are discussed in detail and properly in chater 4.

Reviewer 3 Report

Major Concerns:

In the Methods section, lots of details are missing. For example:

L114- How were the CD4+ cells isolated in blood? It’s not clear.

L233: Discusses a gene-gene interaction analysis but no such analysis is described in the methods.  Were regulatory pathways or genes identified? This section seems incomplete.

L261-264: How was the association analysis between DMGs and DEGs performed?  This was not discussed in the methods.

Regarding the identification of differentially methylated regions: It is not clear why the Fisher test was used since the Methylkit package suggests using this test only when there is one sample per group. It is also not clear how the authors corrected for over-dispersion in the DMR analysis. Did the authors check if age and parity could be affecting methylation levels? As authors pointed out in the introduction (L63) age and other factors may affect methylation. According to Table 1, average parity was a bit higher in the high cortisol group. If these factors are affecting the methylation levels, the authors could account for these covariates in the model using the logistic regression approach, and also correct for the over-dispersion inherent in the sequence count data.

Minor Concerns:

Abstract

The authors did not mention that the analysis were conducted in blood. The title and the abstract give the idea that the research was performed in milk. Please make these points more clear.

L28: Can the authors conclude that the cortisol secretion triggers epigenetic regulation? How do the authors know that it is the cause or consequence of cortisol secretion and is it transgenerational or a physiological programming effect of cortisol?

Introduction

The introduction needs to tell more about the story that authors are trying to investigate.

I would suggest the authors give an explanation as to why they choose to analyze the changes in methylation of blood. It is well explained in the discussion, but this is an important information to the story of this research and would contribute for the reading.

L114: Was DNA collected only from the CD4+ cells?

L117: It is not clear why the two ranges were described: “(150-175 bp and 175-225 bp)”?

L137-138: Please reword- not clear.

L140: Were only CpG were used in this analysis? It is not clear.

L146: This is the first time the term DMG appears in the manuscript. The authors need to define the abbreviation here.

L146: The authors did not specify which levels of GO term they used so I suppose they used all of them. I wonder if the results would be more informative if the authors considered using only higher levels of GO terms? Why not use only biological process and more specific GO levels?

L160-161: Can you be more specific about the Illumina pipeline used? Was it Cassava? What version?

L165: Did the authors filter/ remove genes with extremely low expression?

L176: By variant, I assume you mean high and low cortisol. Please reword. Also, are these mapping percentages reasonable for RRBS data? Could you cite an example? The mapping percentages seem a little low, even for methylation data.

Results

L184: The authors need to define what a CHG and CHH are for readers that are not familiar with these terms.

Figure 2: The legend should be more informative, for example in defining the abbreviations used in the figure.

L191-193: Are they statistically different?

L209: Section 3.4 describing DMRs. There are a lot more analyses that could have been done here. There is relatively little written about changes in methylation in promoter regions for example (despite being described as important in the discussion). Since promoters are thought to be a major area regulated by methylation, it would make sense to investigate these DMR in promoters in more depth.

L213-215: This analysis was not described in the methods.

Figure 5: The legend needs to be improved with more description. For example, what are the grey boxes inside the circle?  Within the figure, the Z score needs numeric scale or statistically significance needs to be added as it’s unclear if this figure provides statistically significant results. Also, q-values should be provided for the enriched BP ontology terms listed.

L222: “These were submitted” What was submitted? Please clarify the text.

Figure 6: What do the red and blue points mean? What is the significance of the genes listed? The figure legend needs considerably more details added.

Figure 7: Many results here are not statistically significant at the reported q value threshold (majority of the results). Also the figure legend needs more details.

L241-244: Did the authors use a threshold of 0.05 or 0.2?

L256-259: Why did the authors list the top 20 and not use the threshold defined in the Methods (q < 0.05)?

L259-261: This would fit better in the discussion, not results.

L261-264: How was the association analysis performed between DMGs and DEGs?

L264-265: What was the main purpose of the DEG analysis? The DEG results and their link with the DMG are not discussed in the paper.

L265- Table 4: q values are required in the table for the DE genes and DMR and more description is needed in the title.

Discussion

L268-283: The authors did a good job of explaining why they chose blood for the analysis. However, I would suggest a quick explanation in the introduction for benefit of the readers, otherwise is hard to know why the discussion goes back and forth from milk to blood.

L282-283: Should this line should be part of the first paragraph?

L304-307: The analyses suggested here are interesting and could be done in this paper (look at location in relation to gene enrichment, network analyses). Why not try?

L311-313: Please, reword this sentence.

L308-381: Several interesting genes were discussed but there were major topics missed in discussion. For example, PDE5 is involved in vasodilation (targeted by pharmaceuticals)- plays a role in heat stress likely due to impact on vasodilation. Also NR5A1 is known to play a critical role in glucocorticoid signaling and tightly tied to cortisol function based on knock out data.

L379-381: Can the authors conclude that this hypo-methylation is a response or the cause of the cortisol level?

L433: The current conclusions do not provide a good summary of the manuscript and could stand improvement.

Typographical issues (suggested changes):

There were many typographical/grammatical issues identified. Here are some examples that should be corrected. The manuscript requires a careful typographical review.

L28: This sentence is missing a “,”. I guess it would be after “livestock”. Please check it.

L100: Please add a space between “analysis.” and “The”.

L101: Please, remove the “.” after the “-20°C”

L104-105: “10 cows with the higher and 10 cows with the lowerSuggest: reword

L129: “reads” Suggest: change to “read”

L175: “millions mapped” Suggest: “million mapped”

L216-218: The text is segmented with 1 sentence by itself in it’s own passage with no apparent need for a new paragraph. Please improve formatting.

L275-276: Reword- multiple grammatical issues.

L282-283: same as L216 above- please correct.

L291: Please, add a “.” after the references “[47,48]”.

L297: Add “,” after “Usually”.

L314: Please reword: “Genes that act as a regulators”. Suggest: “Genes that act as regulators” .

L325: Add “,” after “Interestingly”.

L331: Add “,” after “Normally”.

L355: “found other two important genes”

L387: “Recent evidences”

L424: “Bos taurus” and “Bos indicus” should be in italic.

Round 2

Reviewer 3 Report

I  still have several minor edits that require attention by the authors.

Point 4:  Regarding correction for over-dispersion in the DMR analysis. This is the statistically correct thing to do, regardless  of if another method  gives comparable results, so I would  suggest  adding  to the manuscript that you used this correction and got the same results as used in the current manuscript.

Point 14: L165: The authors say they did filter/ remove genes with extremely low expression.   This  should be mentioned specifically in the methods of the paper with a description of how this  was done.

L241-244: Regarding the multiple testing correction threshold  for the figure, it would be suitable to list a threshold of q = 0.05 or 0.2.  I can understand the potential interest in setting a higher threshold, but some of the results have really high q values (> 0.2) and  are likely false positives.  Respectfully, the two references provided where other authors did something similar were not of high quality and a better standard should be expected for statistical correction of such tests as most authors I  am aware of use q  < 0.10 for ontology enrichment tests.  

Point 26: L261-264: For the association analysis performed between DMGs and DEGs, at what level was significance declared?  q< 0.1, other?  This needs to be described.

One additional typographical correction I note:

L501: "Further researches are also".  I  suggest correcting to:

Further research is also…
